# Physician-Friendly Tool Center Point Calibration Method for Robot-Assisted Puncture Surgery

**DOI:** 10.3390/s21020366

**Published:** 2021-01-07

**Authors:** Leifeng Zhang, Changle Li, Yilun Fan, Xuehe Zhang, Jie Zhao

**Affiliations:** State Key Laboratory of Robotics and Systems, Harbin Institute of Technology, Harbin 150001, China; 18B908094@stu.hit.edu.cn (L.Z.); lichangle@hit.edu.cn (C.L.); fanyilun@hit.edu.cn (Y.F.); jzhao@hit.edu.cn (J.Z.)

**Keywords:** robot TCP calibration, binocular vision, puncture surgery

## Abstract

After each robot end tool replacement, tool center point (TCP) calibration must be performed to achieve precise control of the end tool. This process is also essential for robot-assisted puncture surgery. The purpose of this article is to solve the problems of poor accuracy stability and strong operational dependence in traditional TCP calibration methods and to propose a TCP calibration method that is more suitable for a physician. This paper designs a special binocular vision system and proposes a vision-based TCP calibration algorithm that simultaneously identifies tool center point position (TCPP) and tool center point frame (TCPF). An accuracy test experiment proves that the designed special binocular system has a positioning accuracy of ±0.05 mm. Experimental research shows that the magnitude of the robot configuration set is a key factor affecting the accuracy of TCPP. Accuracy of TCPF is not sensitive to the robot configuration set. Comparison experiments show that the proposed TCP calibration method reduces the time consumption by 82%, improves the accuracy of TCPP by 65% and improves the accuracy of TCPF by 52% compared to the traditional method. Therefore, the method proposed in this article has higher accuracy, better stability, less time consumption and less dependence on the operations than traditional methods, which has a positive effect on the clinical application of high-precision robot-assisted puncture surgery.

## 1. Introduction

Robot-assisted needle insertion technology can improve the accuracy and security of many minimally invasive percutaneous surgeries, such as biopsy and brachytherapy. Usually, a robot-assisted needle insertion system mainly includes a lesion navigation system and a robot [1]. Through a series of coordinate transformations, the robot precisely inserts the needle into the lesion location [2]. Errors from coordinate transformations result in a precise but not accurate needle placement. An important reason is the inaccuracy of the tool center point (TCP) which is often described as the tool center point position (TCPP) and tool center point frame (TCPF) with respect to the end flange frame in robot-assisted puncture surgery. The inaccuracy will result in a fixed offset between the true position of the needle tip and the target position for each penetration. For all robot users, general robot manufacturers provide kinematics calculations only from the base to the end flange instead of the TCP. That means doctors must establish the coordinate transformation from the end flange to the TCP by themselves to adapt to different size of puncture needles and to ensure the safety of the operation. Traditional TCP calibration is to jog a robot manually to approach a sharp tip feature with different robot poses [3]. This procedure is time consuming and highly operator-dependent, since it requires the doctor’s eyes to focus on the sharp tip to ensure that the TCP can reach the same position every time. This method is not conducive to clinical applications and the promotion of precision puncture surgery. Thus, it is necessary to propose a quick, simple and error-controllable TCP calibration method.

Many TCP calibration solutions have been proposed by academia and industrial researchers, with three main approaches, namely, mechanical constraint, laser sensor measuring and vision processing [4].

Mechanical constraint is a common method for traditional manual calibration and is a typical contact method. The most commonly used mechanical constraint is the sharp-tipped tool. The whole process is carried out in two steps. In the first step, TCP is moved by the operator to the tip with more than four different poses to calibration TCPP [5]. In the second step, operator should select at least one points on each axis of the TCP frame taking the TCPP obtained by the first step calibration as the origin and move them to the tip, which is to establish the TCPF. Mizuno proposed a multipoint spherical fitting method that combined least squares to calibrate the TCP [6]. Xiong Shuo combined the coordinate transformation relationship and the least squares matrix to simplify the calculation of the spherical fitting [7]. However, the fact that the TCP cannot completely coincide with the tip leads to low calibration accuracy, and the placement of the TCP is related to the operator’s experience. Using this method for needles means that the operator must gaze fixedly at the needle tip, which is smaller than one millimeter, to ensure that the distance is close enough to obtain a reliable result. The other mechanical constraints reportedly used are a sphere [8] and plane [9]. However, the shortcoming is that the operators must subjectively determine whether the TCP reaches the target location. The DynaCal system also is typical contact calibration equipment that is based on a pull-wire sensor, which has been widely used in industry. However, these contact methods are not suitable for flexible tools such as puncture needles because it is difficult to keep the flexible tool from deforming while pulling the measuring wire attached to the tool.

Therefore, noncontact measurement methods are more suitable for needle calibration. Laser sensors are among the most widely used methods for modern industrial noncontact measurements, due to their high precision and efficiency. ABB’s BullsEye is a famous solution that has been widely used in the calibration of welding torches. BullsEye uses a single laser beam as a line constraint to calculate the robot tool coordinate via a robot motion procedure. Thereafter, Hao Gu proposed a dual laser line TCP calibration method [10]. This method significantly reduces the operating time by more than 50% and easily improves the TCP accuracy to the submillimeter level. These noncontact methods can reach high accuracy and are robust to the environment but are relatively high in cost and require specific trajectories to be preplanned for the robot to ensure accuracy. Vision measurement technology also plays an important role in industry, such as digital image correlation (DIC) [11,12], in surgery, such Optical Coherence Tomography [13,14,15]. Liu et al. [16] proposed an automatic TCP calibration method based on a common binocular vision measurement, which reduced the dependence on operating experience while guaranteeing certain accuracy. However, limited by the accuracy of the binocular system, it is difficult for its calibration accuracy to meet the needs of high precision. Luo et al. [17] combined vision and deep neural networks to achieve adaptive tool calibration. There are also other studies that apply vision to the calibration of other parameters of robots [18,19,20]. Among them, binocular vision is widely favored by researchers. Researchers have summarized the factors that affect the accuracy of the binocular system, including structural parameters and calibration parameters [21,22,23]. The reason is mainly due to the camera’s central perspective model and the binocular measurement principle. In typical binocular systems, the longer the baseline is, the higher the resolution in the direction of the *z*-axis, and at the same time, the view field of the binocular system will be farther away from the camera’s image plane, which will reduce the resolution of the x and y-axes. Thus, it is difficult for a typical binocular system to meet the required accuracy for puncture surgery.

In this paper, we built a high-precision binocular system to achieve high-precision spatial positioning of the puncture needle tip and the direction of its axis. Using the binocular system, we also proposed an calibration algorithm based on the least squares method. The algorithm decouples the TCPP (position of needle tip) and TCPF (direction of needle axis) and identifies them separately in the common process. The largest advantages of our method are that it reduces the difficulty of operations while improving the accuracy. The doctor only needs to control the needle fixed in the robot end flange to enter the measurement space to complete the calibration.

The remainder of this paper is organized into the following sections. In Section 2, we introduce our system’s constitution, the design of vision system, the algorithm for positioning the tip of needle and direction of needle axis and the analysis of TCP calibration algorithm. Section 3 introduced the experimental design of the evaluation of our vision system’s accuracy, the accuracy in different configurations and the comparison of our method and traditional methods. Section 4 analyzes the results of all experiments and suggestions for calibration using the method proposed in this article are given. Section 5 concludes that our proposed method can achieve a higher precision puncture and has higher clinical application value.

## 2. Materials and Methods

### 2.1. System Constitution

The experiment system is composed of a collaborative robot, binocular vision system and PC software. The system configuration is shown in Figure 1. In this system, we adopt a 6 DOF collaborative robot offered by UNIVERSAL ROBOTS Co., Ltd. (Shanghai, China), model UR5. A biopsy needle (provided by Bard Peripheral Vascular, Inc, MN1413) is mounted on a simple fixture attached to the end of the flange. The binocular vision system is formed by two industrial CCD cameras offered by DAHENG IMAGING Co., Ltd. (Beijing, China). The camera model is MER-1810-21U3C and is equipped with a lens of model M1224-MPW2. The resolution of each camera is 4912 (H) × 3684 (V). The focus of the lens is 12 mm. The system applies diffuse bright-field back light illumination to improve the image contrast, thereby improving the measurement accuracy. In addition, this illumination design can help the camera to obtain a clear image with a small aperture. The advantage of a small aperture is that the camera’s depth of field is larger, and a larger measurement space generated can be used to capture the movement of the needle tip, avoiding the out-of-focus caused by the movement of the needle tip in the measurement space.

While the system is working, the operator manually jogs the robotic arm to place the puncture needle into the stereo vision measurement space. The robot controller sends the end flange position to a personal computer (PC) in real time by an Ethernet cable. The images are captured by two cameras through a universal serial bus (USB) cable to the PC. The position of the needle tip in the measurement space coordinate system can be calculated by an image processing algorithm running on the PC. All of the algorithms are written in C++ based on two external libraries, Halcon and Eigen.

### 2.2. Binocular System Design and Image Processing

The core task of the binocular system is to accurately position the needle. Binocular stereo vision acquires three-dimensional information on the objects based on the principle of parallax. Therefore, parallax is an important factor that affects the resolution of the binocular system. The resolution of a traditional binocular system is mainly affected by the baseline length and the measurement distance without considering the camera’s own parameters. However, longer baselines lead to longer minimum measurement distances, which is not conductive to accurate measurements. According to the TCP calibration requirements of the robot with a needle as the end tool, we designed a convergent binocular system. The increase in the angle between the optical axes of the two cameras improves the resolution and reduces the size of the uncertain area [24].

The essence of vision-based measurement technology is the correspondence between spatially uncertain regions and the vision system’s pixels. Figure 2 shows the correspondence between the binocular system’s pixels and the spatially uncertain area using a simplified model of a binocular vision system. This model projects the uncertain region onto the horizontal plane assuming that the two cameras are placed at the same height. According to the principle of imaging, the two boundaries of a single pixel correspond to two rays in space, as shown by the green line in the figure, thereby forming a quadrangular uncertain region (green area). Obviously, ω will significantly affect the size of the uncertainty area.

We analyzed the size of the uncertain region at different optical axis angles through numerical calculations. We fixed the length of the baseline and calculated Δx, Δz and $\sqrt{\Delta x^{2}+\Delta z^{2}}$ of uncertain regions at the angle ω of 0°, 15°, 30°, 45°, 60° and 75°. The contour in Figure 3 shows this result. Overall, as ω increases, Δx and $\sqrt{\Delta x^{2}+\Delta z^{2}}$ show a trend of decreasing first and then increasing. Δz decreases as ω increases.

Since the calibration process does not use all of the view field, we chose the nearest circular area with a radius of 20 mm as the analysis object and analyzed the mean and variance of Δx, Δz and $\sqrt{\Delta x^{2}+\Delta z^{2}}$. The results are shown in Figure 4. We can see that the mean of Δx and $\sqrt{\Delta x^{2}+\Delta z^{2}}$ decreases first and then increases as ω increases, and it reaches a minimum when ω is 45°. The mean of Δz decreases with an increase in ω. In general, the fluctuation of Δx is relatively mild. Δz and $\sqrt{\Delta x^{2}+\Delta z^{2}}$ are drastically reduced when ω goes from 0° to 30°. To facilitate a comparison of the mean and variance, we calculated the negative logarithm of all of the variances. The negative logarithm of the variance of Δx and $\sqrt{\Delta x^{2}+\Delta z^{2}}$ increases first and then decreases as ω increases, and it reaches a maximum when ω is 45°. The negative logarithm of variance of Δz increases with increased ω. It can be seen from the variance that the accuracy of the visual system is more stable when ω is 45°. In summary, when ω is 45°, the vision system has better performance for the positioning accuracy of internal points in the circular area, theoretically.

Based on the above numerical simulation results and analysis, we designed an orthogonal binocular system, as shown in Figure 5. Such a configuration is not conducive to the matching of feature points because in the most extreme cases the images of the two cameras are completely different. However, this will not affect the algorithm introduced in Section 2.3. The left camera and right camera are fixed on two cross slides. The relative positions of the two cameras can be controlled by the two cross slides to ensure that both cameras have a clear field of view. Due to the use of a small aperture to increase the depth of the scene, the system adds four light sources to assist the imaging. Two ring lights are fixed in front of the lens for the calibration of the internal and external parameters of the vision system. Two back lights are fixed on the two cameras to enhance the contrast of the image and to accurately extract the needle tip position. The four lights are controlled by the lighting controller. All of the devices are fixed on an optical shock absorption platform.

### 2.3. Positioning Needle

As is well known, the binocular system must be calibrated for internal and external parameters before use. This process has been systematic, and we used HALCON’s toolbox and a circular marking calibration plate to calibrate the internal and external parameters of the system. Since our goal is positioning the needle tip rather than three-dimensional reconstruction of an object, we propose a simple algorithm for accurately acquiring position of needle tip and direction of needle axis in our vision system.

The images processing is described as follows to get pixel coordinate of the needle:
(a)Take gray pictures when no objects are placed in the binocular system and record separately as $I_{L}\left(x,y\right)$ and $I_{R}\left(x,y\right)$;(b)Control the robot moving the needle tip to different positions within the measurement range of the binocular system and take pictures $I_{L}^{i}\left(x,y\right)$ and $I_{R}^{i}\left(x,y\right)$ (i=1,2,3⋯n);(c)Subtract the image that contains the needle tip from the image that corresponds to the initial state of the camera without the needle tip using formula $G_{L\left(R\right)}^{i}\left(x,y\right)=\left(I_{L(R)}^{i}\left(x,y\right)-I_{L}\left(x,y\right)\right)+128$. The gray value of pixels less than 0 is truncated to 0 and greater than 255 is truncated to 255;(d)Select the pixels from $G_{L\left(R\right)}^{i}\left(x,y\right)$ whose gray values fulfill the condition $0\leq G_{L(R)}^{i}\left(x,y\right)\leq 100$ based on the experience of the experimental;(e)The resulting image will contain the needle and partial noise. We calculate the size of all of the connected domains in the image and keep the largest connected domain. Then correct image distortion. Then, using a circular structure with a radius of 5 pixels, we perform a morphological opening operation on the image to smooth the outline of the needle.(f)Calculate the maximum circumscribed rectangle of the needle in the image and calculate the coordinates of all pixels where the short side intersects the boundary of the needle. Take the average of all intersection coordinates as the pixel coordinates of the needle tip.(g)Fit the edge of the needle with a polygon. Extract the two longest straight lines as input for calculating the needle direction.


Figure 6 shows a sample of the image processing.

The needle imaging on the two cameras is shown in Figure 7. In this article we take the frame of left camera as the world frame. The 3 × 4 matrix P maps needle tip $X={\left[\begin{array}{cccc} X & Y & Z & 1 \end{array}\right]}^{⊤}$ as a point from 3D space to the 2D image space, $x^{⊤}=\left[\begin{array}{ccc} x & y & 1 \end{array}\right]$, via perspective projection up to a scale w.
(1)$$w_{L}x_{L}=\underset{P_{L}}{\underset{︸}{\left[\begin{array}{ccc} \alpha_{yL} & 0 & x_{0L}\\ 0 & \alpha_{yL} & y_{0L}\\ 0 & 0 & 1 \end{array}\right]\left[I|0\right]}}X$$
(2)$$w_{R}x_{R}=\underset{P_{R}}{\underset{︸}{\left[\begin{array}{ccc} \alpha_{xR} & 0 & x_{0R}\\ 0 & \alpha_{yR} & y_{0R}\\ 0 & 0 & 1 \end{array}\right]\left[R|t\right]}}X$$
where the subscripts L and R represent the left camera and the right camera, respectively. $\alpha_{x}=m_{x}f$ and $\alpha_{y}=m_{y}f$ represent the focal length f of the camera in terms of pixel dimensions in the x and y direction, respectively. $m_{x}$ and $m_{y}$ is the number of pixels per unit distance in image coordinate. $x_{0}$ and $y_{0}$ is the principal point in terms of pixel dimensions. Because the left camera is set as the world frame of the vision system, the external parameter of the left camera is [I|0], which means there is no rotation and translation. The external parameter of the right camera is [R|t], where R and t represent rotation matrix and translation vector relative to the left camera.

Needle tip position in 3D space $X={\left[\begin{array}{cccc} X & Y & Z & 1 \end{array}\right]}^{⊤}$ calculate by Formula (3) using least square method assuming that the needle tip is not at infinity [25].
(3)$$\left[\begin{array}{c} x_{L}p_{L}^{3⊤}-p_{L}^{1⊤}\\ y_{L}p_{L}^{3⊤}-p_{L}^{2⊤}\\ x_{R}p_{R}^{3⊤}-p_{R}^{1⊤}\\ x_{R}p_{R}^{3⊤}-p_{R}^{2⊤} \end{array}\right]\cdot \left[\begin{array}{c} X\\ Y\\ Z\\ 1 \end{array}\right]=0$$
where $p_{}^{i⊤}$ is rows of p.

We considered TCPF calibration as the direction calibration problem of straight homogeneous generalized cylinders (SHGC) [26,27,28,29,30]. We used an analytical geometric method that combines the law of SHGC projection imaging to estimate it. The projection of the cylinder on the each image plane will always include two straight lines such as $l_{L}^{+}$ and $l_{L}^{-}$ as shown in Figure 7. They are formed by the intersection of the image plane and the tangent plane of the cylinder which pass through the camera’s optical center. We represented these tangent planes by the normal vectors $n_{L}^{+}$
$n_{L}^{-}$
$n_{R}^{+}$ and $n_{R}^{-}$. The direction of the normal vectors is given by
(4)$$n=\frac{P^{⊤}l}{\left\|P^{⊤}l\right\|}$$
where l can be $l_{L}^{+}$
$l_{L}^{-}$
$l_{R}^{+}$ and $l_{R}^{-}$ given in Hesse normal form on the image plane and n calculated corresponds to $n_{L}^{+}$
$n_{L}^{-}$
$n_{R}^{+}$ and $n_{R}^{-}$ using $P_{L}^{⊤}$ and $P_{R}^{⊤}$.

The axis of the needle must be on the symmetry plane $n_{L}$ and $n_{R}$ with
(5)$$n_{L(R)}=n_{L(R)}^{+}+\left(\frac{n_{L(R)}^{+}\cdot n_{L(R)}^{-}}{\left\|n_{L(R)}^{+}\cdot n_{L(R)}^{-}\right\|}\right)n_{L(R)}^{-}$$

$\frac{n_{L(R)}^{+}\cdot n_{L(R)}^{-}}{\left\|n_{L(R)}^{+}\cdot n_{L(R)}^{-}\right\|}$ is a corrector factor to ensure that $n_{L(R)}$ is perpendicular instead of parallel to the symmetrical plane.

Then the direction of needle axis can be calculated by Formula (6).
(6)$$l_{Axis}=\frac{n_{L}\times \left(R\cdot n_{R}\right)}{\left\|n_{L}\times n_{R}\right\|}$$

### 2.4. TCP Calibration Algorithm

TCP calibration is to obtain the actual position and frame of the tool center point using external measurements and fitting algorithms. The essence of this problem is to solve the problem of AX = B, and many researchers have already given solutions [31,32,33]. With regard to a surgical assistant robot, on the one hand, it requires high accuracy of the TCP, and on the other hand, it must reduce the difficulty of the doctors’ operations using the robot. As shown in Figure 8, frame {**B**} is the robot’s base frame. Frame {**E**} is the end flange frame. Frame {**V**} is the binocular vision system frame.

For the needle, we do not care about the rotation of the needle tip frame with respect to the needle axis. We only need to align the z-axis of the tip frame with the axis of needle. So the position and pose are considered separately as the position vector of the needle tip ($P_{Needle}^{E}$) and the direction vector of the needle axis ($l_{Axis}^{E}$) in frame {**E**}. According to the forward kinematics of the robot, any vector $P_{Needle}^{E}$ satisfies the following formula in the frame {**E**}.
(7)$$T_{V}^{B}\cdot {\left[\begin{array}{cc} P_{i}^{V} & 1 \end{array}\right]}^{⊤}=T_{Ei}^{B}\cdot {\left[\begin{array}{cc} P_{Needle}^{E} & 1 \end{array}\right]}^{⊤}\;\left(i=0,1,2,3\cdots ,n\right)$$
In this function, $T_{V}^{B}=\left[\begin{array}{cc} R_{V}^{B} & t_{V}^{B}\\ 0 & 1 \end{array}\right]$ is the homogeneous transformation from {**B**} to {**V**}, and it is an unknown constant during the calibration. $\left[\;\begin{array}{cc} P_{i}^{V} & 1 \end{array}\right]{}^{Τ}={\left[\begin{array}{cccc} x_{i}^{V} & y_{i}^{V} & z_{i}^{V} & 1 \end{array}\right]}^{⊤}$ is the needle tip position in frame {**V**} (as $X={\left[\begin{array}{cccc} X & Y & Z & 1 \end{array}\right]}^{⊤}$ remove the homogeneous factor in Section 2.3). $T_{Ei}^{B}=\left[\begin{array}{cc} R_{Ei}^{B} & t_{Ei}^{B}\\ 0 & 1 \end{array}\right]$ is the homogeneous transformation from {**B**} to {**E_i_**}, and we can obtain it from the robot controller. $\left[\begin{array}{cc} P_{Needle}^{E} & 1 \end{array}\right]{}^{Τ}={\left[\begin{array}{cccc} x^{E} & y^{E} & z^{E} & 1 \end{array}\right]}^{Τ}$ is the position vector of the needle tip in frame {**E}**. We expand Formula (7) to obtain Formula (8). Obviously, there are two unknown vectors and one unknown matrix in the equation. It is impossible to directly obtain $P_{Needle}^{E}$.
(8)$$R_{V}^{B}\cdot P_{i}^{V}+t_{V}^{B}=R_{Ei}^{B}\cdot P_{Needle}^{E}+t_{Ei}^{B}\;\left(i=0,1,2,3\cdots ,n\right)$$

Moving *n* + 1 positions while keeping a fixed rotation of the robot’s end effector, a series of Equation (8) can be obtained. By calculating the difference between every equations and the first equation, we obtain n equations as in Equation (9), which contains only one unknown matrix $R_{V}^{B}$. Because the end flange makes only translational motions and frame {**V**} remains relatively fixed with frame {**B**}, $R_{V}^{B}$ and $R_{Ei}^{B}\cdot P^{E}$ are constant.
(9)$$R_{V}^{B}\cdot \left(P_{i}^{V}-P_{i+1}^{V}\right)=t_{Ei}^{B}-t_{E0}^{B}\;\left(i=1,2,3\cdots ,n\right)$$

Next, we use singular value decomposition (SVD) to obtain $R_{V}^{B}$.

Then we manually control the end flange to move to m + 1 (four or more) points with translation and rotation. We should attempt to make a difference in the posture between each point large enough. The same as above, by calculating the difference between every equations and the first equation, we obtain m equations, which contains one unknown vector $P_{Needle}^{E}$. We also use SVD to find $P_{Needle}^{E}$.
(10)$$\left(R_{Ei}^{B}-R_{E0}^{B}\right)\cdot P_{Needle}^{E}=R_{V}^{B}\cdot \left(P_{i}^{V}-P_{0}^{V}\;\right)-\left(t_{Ei}^{B}-t_{E0}^{B}\right)\;\left(i=1,2,3\cdots ,m\right)$$

The direction of needle axis $l_{Axis}^{E}$ in end flange frame always follow Formula (11). We also use SVD decomposition to get $l_{Axis}^{E}$.
(11)$$R_{B}^{V}\cdot R_{Ei}^{B}\cdot l_{Axis}^{E}=l_{i}^{V}\left(i=0,1,2,3\cdots ,m\right)$$
where $l_{Axis}^{E}$ represents the direction of needle axis in end flange frame, $l_{i}^{V}$ represents the direction of needle axis in vision frame (as $l_{Axis}^{}$ in Section 2.3).

In order to facilitate the robot controller to directly control the needle, the homogeneous transformation matrix between the end flange and the needle tip must be calculated. Although some researches have pointed out that the asymmetry of the side-bevel needle will cause the puncture trajectory to be deflected, but this is only limited to the muscle-rich positions such as head and neck surgery [34]. More applications, including the chest, abdomen and prostate, where there are more cavities, the deflection is not obvious, and the use of diamond-tip needles also reduces the deflection. So, without considering the rotation of the needle tip frame with respect to the needle axis, rotation matrix is simplified to the following.
(12)$$\underset{R_{Needle}^{E}}{\underset{︸}{\left[\begin{array}{ccc} \cos\beta & 0 & \sin\beta \\ 0 & 1 & 0\\ -\sin\beta & 0 & \cos\beta \end{array}\right]\left[\begin{array}{ccc} 1 & 0 & 0\\ 0 & \cos\gamma & -\sin\gamma \\ 0 & \sin\gamma & \cos\gamma \end{array}\right]}}\left[\begin{array}{c} 0\\ 0\\ 1 \end{array}\right]=l_{Axis}^{E}$$

Finally, the homogeneous conversion matrix $\left[\begin{array}{cc} R_{Needle}^{E} & P_{Needle}^{E}\\ 0 & 1 \end{array}\right]$ from the end flange to the tip of the needle is obtained.

In the second step of TCP calibration, the difference between all pose needs to be as large as possible. The reason is that the least squares estimate of Formulas (10) and (11) is related to this difference. In order to facilitate the following analysis, we simplify (10) and (11) to A⋅X=B. The bound of the estimation of X is as follows [35]:(13)$${\left\|\tilde{X}-X\right\|}_{2}\leq \frac{\varepsilon \cdot \kappa_{2}\left(A\right)}{1-\varepsilon \cdot \kappa_{2}\left(A\right)}\left(2+\left(\kappa_{2}\left(A\right)+1\right)\frac{{\left\|r\right\|}_{2}}{{\left\|A\right\|}_{2}{\left\|X\right\|}_{2}}\right){\left\|X\right\|}_{2}$$
where $\varepsilon \equiv \max\left(\frac{{\left\|\delta A\right\|}_{2}}{{\left\|A\right\|}_{2}},\frac{{\left\|\delta B\right\|}_{2}}{{\left\|B\right\|}_{2}}\right)$, $\kappa_{2}\left(A\right)=\frac{\sigma_{\max}\left(A\right)}{\sigma_{\min}\left(A\right)}$ means the condition number of A, r=AX−B. Obviously the larger $\kappa_{2}\left(A\right)$ and ε, the larger the error ${\left\|\tilde{X}-X\right\|}_{2}$. The difference between all poses determine $\kappa_{2}\left(A\right)$ and ε. Obviously, the binocular visual system we designed limits this difference and may reduce the calibration accuracy. Therefore, the Section 3.2 designed an experiment to analyze the impact of the reduction on calibration accuracy.

## 3. Experimental Design

### 3.1. Accuracy Evaluation of Vision System

To test the actual accuracy of the vision system, we build a test system, as shown in Figure 9. A laser tracker (Leica AT901 Laser Tracker) is used to measure the accuracy of the vision system, which accuracy is ±15 μm + 6 μm/m for position. The spherically reflector of the laser tracker is fixed at the end of the robot. The distance between the reflector and the laser tracker is about 2 m. Thus, through the movement of the reflector, we can obtain a precise change in the needle tip position while keeping the posture of the robot end flange fixed. To evaluate the positioning accuracy of the vision system in all directions, we manually controlled the robot to move 15 intervals at 2 mm on the xy*z*-axis of the robot base frame separately. We did not use the conversion matrix of the vision system frame and the robot base frame to convert the robot’s movement on the xyz axis to the xyz axis of the vision system, because the inaccuracy of the conversion matrix will introduce additional errors. This will not affect the final result because the error of the vision system will not change due to the change of the reference frame. The coordinates of the needle tip in the vision system are calculated by the proposed method. However, the coordinates of the needle tip cannot be calculated in the laser tracker frame and robot base frame. The puncture needle, robot end flange and spherically mounted reflector are relatively fixed, and therefore, their movement has a theoretical relationship, as shown in Equation (14).
(14)$$\{\begin{array}{c} {\left\|P_{i}^{V}-P_{i+1}^{V}\right\|}_{needle}={\left\|P_{i}^{Laser}-P_{i+1}^{Laser}\right\|}_{reflector}={\left\|P_{i}^{B}-P_{i+1}^{B}\right\|}_{EndFlange}\\ P_{i}^{V}=\left[\begin{array}{c} x_{i}^{V}\\ y_{i}^{V}\\ z_{i}^{V} \end{array}\right],P_{i}^{Laser}=\left[\begin{array}{c} x_{i}^{Laser}\\ y_{i}^{Laser}\\ z_{i}^{Laser} \end{array}\right],P_{i}^{B}=\left[\begin{array}{c} x_{i}^{B}\\ y_{i}^{B}\\ z_{i}^{B} \end{array}\right] \end{array}$$

In the above equation, ${\left\|P_{i}^{V}-P_{i+1}^{V}\right\|}_{needle}$, ${\left\|P_{i}^{Laser}-P_{i+1}^{Laser}\right\|}_{reflector}$ and ${\left\|P_{i}^{B}-P_{i+1}^{B}\right\|}_{EndFlange}$ are the Euclidean distance of the needle tip in the vision frame, the Euclidean distance of the reflector in the laser tracker frame and the Euclidean distance of the end flange in the robot base frame, respectively. The laser tracker is used as a standard to evaluate the accuracy of the vision system. The error is defined in Formula (15).
(15)$$P_{error}={\left\|P_{i}^{V}-P_{i+1}^{V}\right\|}_{needle}-{\left\|P_{i}^{Laser}-P_{i+1}^{Laser}\right\|}_{reflector}$$

### 3.2. TCP Calibration Accuracy under Different Configurations

The analysis in Section 2.4 shows that different configurations will change the upper boundary of calibration accuracy. In order to separate the influence of $\kappa_{2}\left(A\right)$ and ε on the accuracy of calibration, the following experiment is designed.

We considered that $\kappa_{2}\left(A\right)$ represents the uniformity of the configuration set in spatial and ε represents the amplitude of the configuration set. Based on experience, we designed 5 configurations for TCP calibration as Figure 10 shows to separate the influence of $\kappa_{2}\left(A\right)$ and ε. The angle between axis of 1, 3, and plane I are both *θ*. The angle between axis of 2, 4, and plane Π are both *θ*. Plane I and plane Π is perpendicular to horizontal plane (plane III). Although all the axis of needle in Figure 10 intersect at one point only to illustrate the experiment, it is not actually necessary for all the axis to intersect at one point. The experiment was divided into two groups and Table 1 described the details:
*θ* was fixed, and ϕ was changed. This configuration would change $\kappa_{2}\left(A\right)$.ϕ was fixed, and *θ* was changed. This configuration would change ε.


A relatively accurate TCP including $P_{real}^{E}$ and $l_{real}^{E}$ were obtained through repeated calibrations assuming that the error between the calibrated TCP and the real TCP obeys zero mean normal distribution. The errors of TCPP ($P_{Needle}^{E}$) and TCPF ($l_{Axis}^{E}$) are evaluated by Formulas (16) and (17), where TCPP and TCPF are the average of 20 serious calibrations using traditional methods.
(16)$$Distance=\left\|P_{real}^{E}-P_{Needle}^{E}\right\|$$
(17)$$Angle=\arccos\left(l_{real}^{E}\cdot l_{Axis}^{E}\right)$$

### 3.3. Comparison of the TCP Calibration with Traditional Methods

The time consumption and accuracy of the calibration are the key criteria for evaluating the practicability. To evaluate the practicability of the new TCP calibration method, a comparative experiment between the traditional method and the proposed method was taken. Using our proposed method, in each experiment, first the operator controls the tool of robot to enter the vision system’s view filed and to move two points randomly along each direction of xyz under a fixed pose to calculate $R_{V}^{B}$. Then, the operator arbitrarily gives the robot five different postures and controls the needle tip to enter the vision system’s view field to calculate $l_{Axis}^{E}$ and $P_{Needle}^{E}$ by the method described in Section 2. The five postures should follow the suggestions given in the conclusion. The error is calculated by Formulas (16) and (17). We repeated the above experiment and also used the traditional method to experiment 15 times. We recorded the time used each time separately.

## 4. Results and Discussion

### 4.1. Accuracy Evaluation of Vision System

The calibrated internal and external parameters are listed in Table 2. The reprojection error is 0.28 pixels. The calibration results meet the experimental requirements.

The accuracy of the vision system is shown in Figure 11. In Figure 11, xyz represents the direction of the robot’s base coordinate and the errors are calculated by Formula (15). It can be seen from the results that all of the errors fluctuate in the range ±0.05 mm and the maximum error reach to 0.041 mm, which is lower than the absolute positioning error of a general cooperative robot. The actual accuracy of the vision system is worse than the simulation results, as mentioned in Section 2.2. On the one hand, this finding is due to the inaccuracy of the internal and external parameters, and on the other hand, the reason is that the needle tip obtained by the image processing algorithm in the two cameras is not the identical physical point. However, the error of ±0.05 mm is still lower than the spatial resolution error of the human eye, in the least distance for distinct vision. This finding means that our vision system using the method proposed can accurately measure the spatial movement of the robot TCP, specifically with the puncture needle as the end tool.

### 4.2. TCP Calibration Accuracy under Different Configurations

The error of needle tip position and needle axis direction is as shown in Figure 12 and Figure 13, which are calculated by Formulas (16) and (17), respectively, when θ was fixed, and ϕ was changed. The increase in ϕ causes the condition number to decrease sharply, especially when ϕ is relatively small. The error of needle tip position has the same trend as condition number, which is consistent with the theoretical analysis of the Section 2. When ϕ is greater than 30°, the error is stable at 0.1 mm. This is because the difference operation in the coefficient matrix in Formula (10) causes ϕ to affect the condition number of the coefficient matrix, which makes it more sensitive to noise and eventually leads to a very large TCPP error when ϕ is small.

Figure 13 shows that the change of ϕ is not sensitive to the condition number of the coefficient matrix of the Formula (11). The increase in ϕ causes the condition number to decrease slowly, because the coefficient matrix of Formula (11) does not involve the difference of the rotation matrix. The needle axis direction error is stable at about 1°, and the maximum can reach 2°. The difference from Formula (10) is that the coefficient matrix of Formula (11) does not involve the difference operation between different poses and its coefficient matrix is all composed of orthonormal matrices, so its condition number does not change significantly, which ultimately leads to the calibration of TCFF not affected by ϕ influences.

The error of TCPP and TCPF are as shown in Figure 14 and Figure 15, which are calculated by Formulas (16) and (17), respectively, when ϕ was fixed, and θ was changed. Under the control of the experiment, the condition number remains unchanged. But the increase of θ also causes the error of TCPP decreases. It is worth noting that although the increase in θ also leads to a decrease in the error, it does not have a large impact on the error than ϕ. This is consistent with theoretical expectations as Section 2.4. The increase in θ leads to the increase in ${\left\|A\right\|}_{2}$ and ${\left\|B\right\|}_{2}$ in Formula (10), which leads to the decrease in ε, which improves the calibration accuracy. Figure 15 shows that the change of ϕ is also not sensitive to the condition number of the coefficient matrix of the Formula (11) and the needle axis direction error. Because B is always the unit vector in Formula (11) and its size is fixed, A is always composed of orthonormal matrices and its modulus is also unchanged.

### 4.3. Comparison of TCP Calibration with Traditional Methods

The time consumption of each experiment is shown in Figure 16. The average calibration time is 482 s and 88 s. The proposed TCP calibration method reduces the time consumption by 82%. The time consumption of the traditional methods varies greatly: the longest time is over 600 s, and the shortest time is close to 400 s. However, the time consumption of proposed method in this paper fluctuates around the average. On the one hand we found that the most time-consuming process is to control the tip of the needle as it reaches the same position every time using traditional methods. Due to the different operability of the robot in different postures, the time consumption varies greatly in each experiment. However, our proposed method only ensures that the needle tip be within the measurement space of the vision system, which is very friendly for physicians and is time-saving. On the other hand, the second step of the proposed method get the TCP position and TCP frame simultaneously. The proposed method gives an operation-independent workflow that is the key to reducing the time consumption.

The error of TCPP and TCPF of each experiment are shown in Figure 17 and Figure 18, which are calculated by Formulas (16) and (17), respectively. The average error of TCPP in 15 experiments using the proposed method is 0.11 mm, which is lower than the result of the traditional method of 0.32 mm. The proposed TCP calibration method reduces the error by 65%. It can be clearly seen that the proposed method has less data fluctuation. This finding shows that the proposed method has not only high accuracy but also high stability. Although occasionally the TCFF error is larger than the traditional method, the average accuracy is still improved by 52%. This is because the diameter of the needle is small and the operator’s habit leads to the same distance between the needle surface and the reference point every time. Therefore, the traditional method in the needle axis direction calibration is also more likely to achieve higher accuracy.

## 5. Conclusions

This paper has presented a novel approach for robot TCP calibration based on a special binocular vision system specifically with the puncture needle as the end tool. Compared with existing solutions, it is easier and faster to operate for physicians and more stable, and its accurate results can ensure the safety of surgery.

In this paper, we designed a binocular vision system as an extended measurement device to tackle the issue of TCP calibration due to the low rigidity and small size when the puncture needle is used as the end tool of the robot. Through the simulation of a simplified model, the vision system that we designed was proven to have a higher spatial resolution than the traditional binocular system and is more suitable for TCP calibration. In addition, we also proposed a TCP calibration algorithm combined with the designed visual system and analyzed the key factors affecting accuracy. Experiments have been conducted with a system that consists of a robot arm, laser tracker and two cameras. The experiment on the visual system’s accuracy evaluation confirmed that the visual system and the corresponding method we proposed can detect the spatial position changes in the TCP (needle tip) with high accuracy. The experiment in different configuration indicates the uniformity of configuration is the main influence factors. Using the binocular vision system only reduces the magnitude of the configuration and the results indicate the reduction barely influence the accuracy of TCP. In terms of TCPF, configuration does not influence its accuracy. The comparison experiment confirmed that the proposed method takes less time and has higher accuracy than the traditional method. A recommended calibration suggestion is to give priority to guarantee ϕ when the space is limited, try to make it close to 90°. In theory, θ should be as close to 90° as possible without collision, but θ should be guaranteed to exceed 30 degrees under limited space.

To sum up, this paper proposed a new TCP calibration method, which is operation-independent and physician-friendly under the premise of ensuring accuracy. It provides a necessary solution for robotic assisted puncture operation to enter the clinic. However, the method not only is suitable for using in the operating room environment but can also can be extended to the production of some electronic products that require precise operations. This method will accelerate the arrangement of production lines and improve product quality, which will promote the automation of robot processing and manufacturing. Future work will focus on TCP calibration for more general end tools.

## Figures and Tables

**Figure 1 sensors-21-00366-f001:**
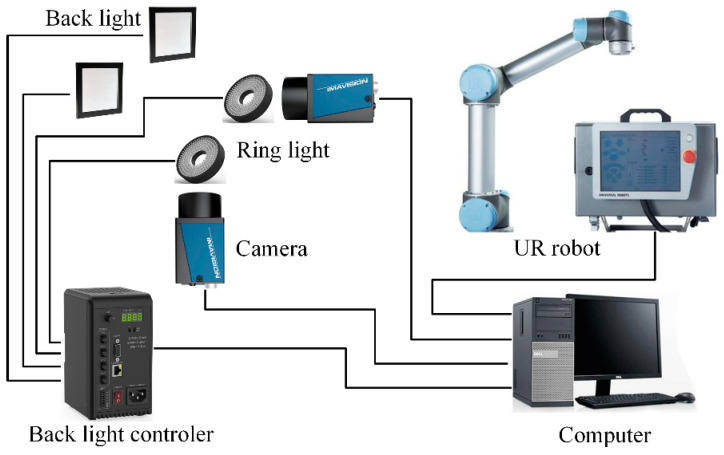
System configuration.

**Figure 2 sensors-21-00366-f002:**
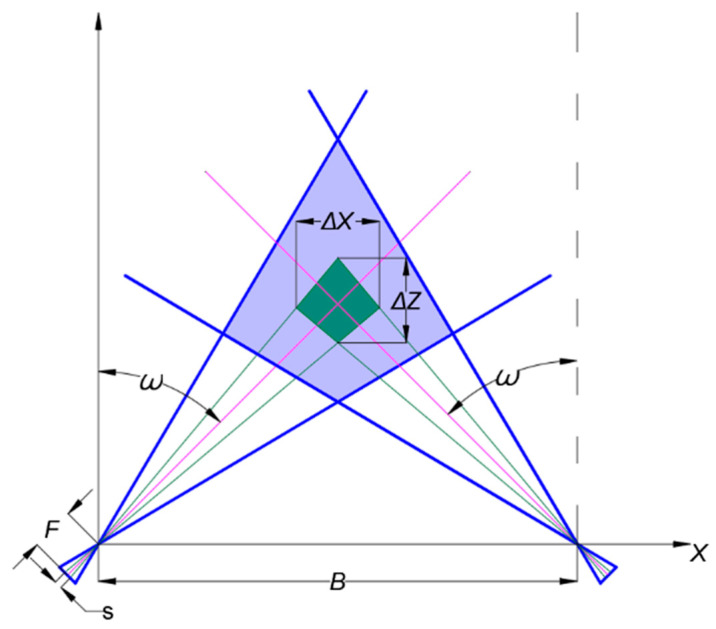
Principles of uncertain regions. (Δx—the size of the uncertain area in X axis, Δz—the size of the uncertain area in Z axis, ω—the angle between the optical axis of the camera and the Z axis, F—the focal length of the lens, s—the size of a single pixel in the simplified model, B—the length of the baseline of the binocular system).

**Figure 3 sensors-21-00366-f003:**
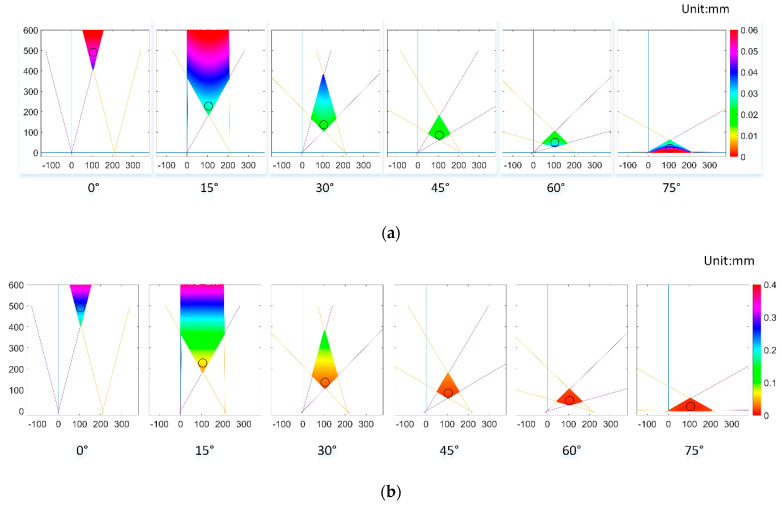
Contour of uncertain region size. (**a**). Δx of uncertain region. (**b**). Δz of uncertain region (**c**). $\sqrt{\Delta x^{2}+\Delta z^{2}}$ of uncertain region.

**Figure 4 sensors-21-00366-f004:**
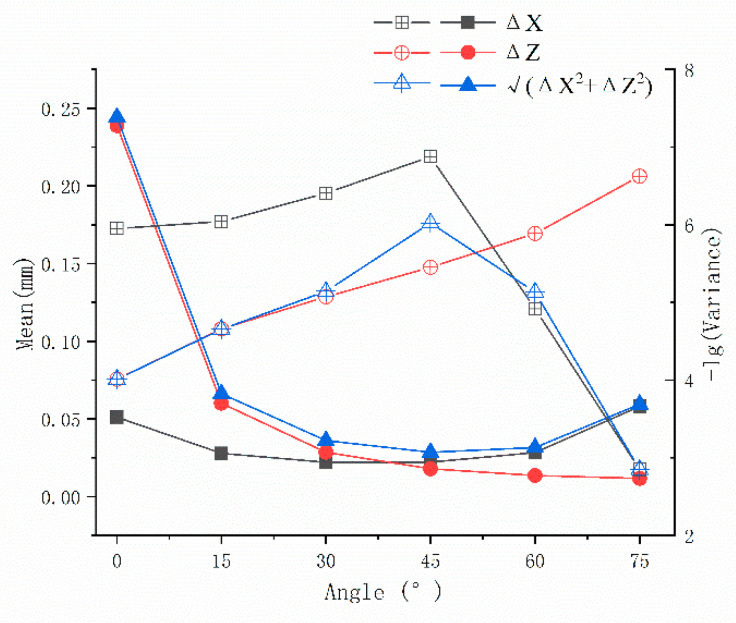
Mean and variance of the uncertain region size in the nearest circular area.

**Figure 5 sensors-21-00366-f005:**
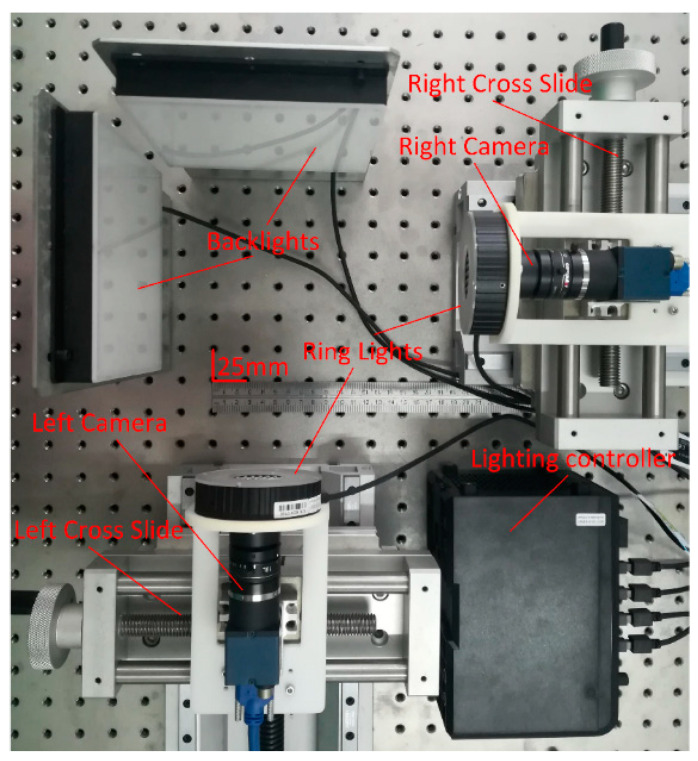
Vision system.

**Figure 6 sensors-21-00366-f006:**
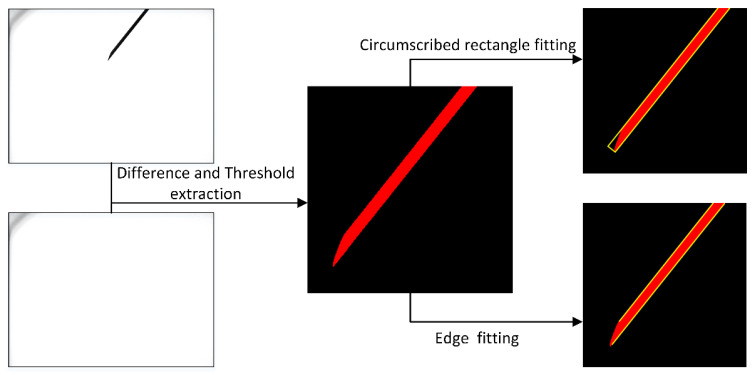
Sample of image processing.

**Figure 7 sensors-21-00366-f007:**
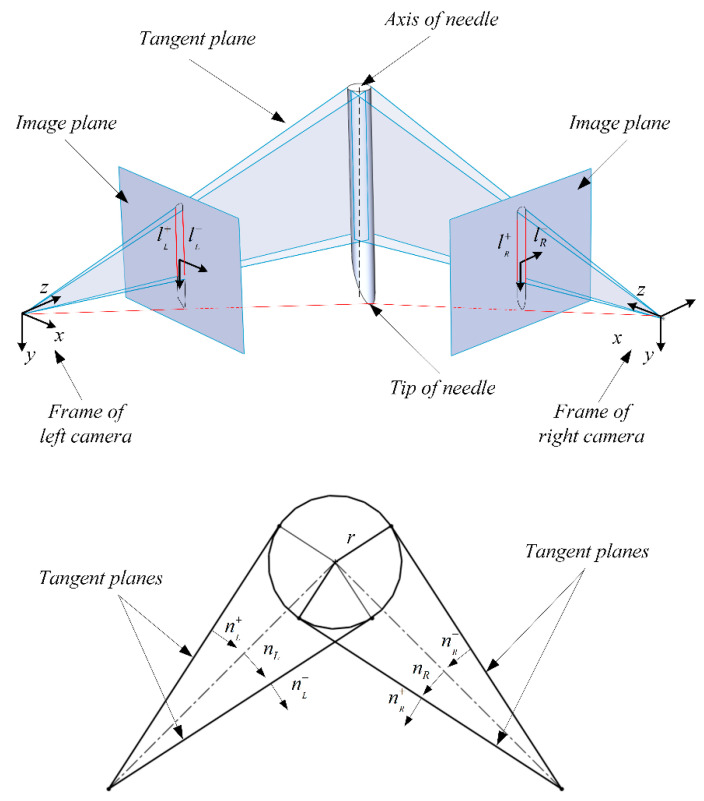
Imaging model.

**Figure 8 sensors-21-00366-f008:**
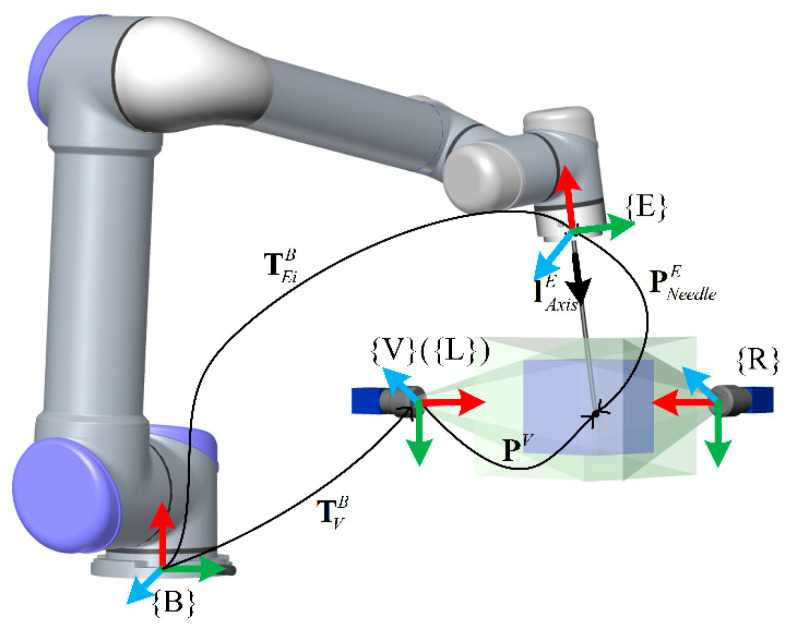
Description of the coordinate system.

**Figure 9 sensors-21-00366-f009:**
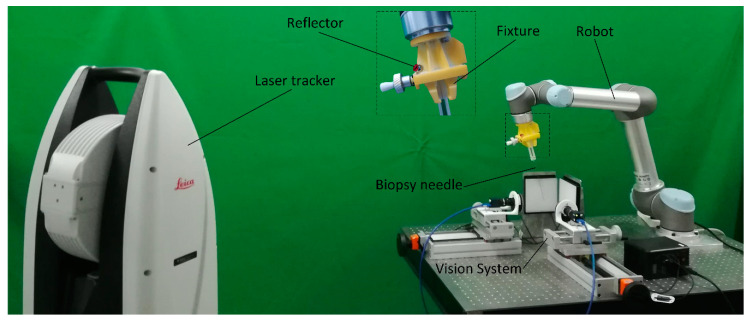
Accuracy testing experiment.

**Figure 10 sensors-21-00366-f010:**
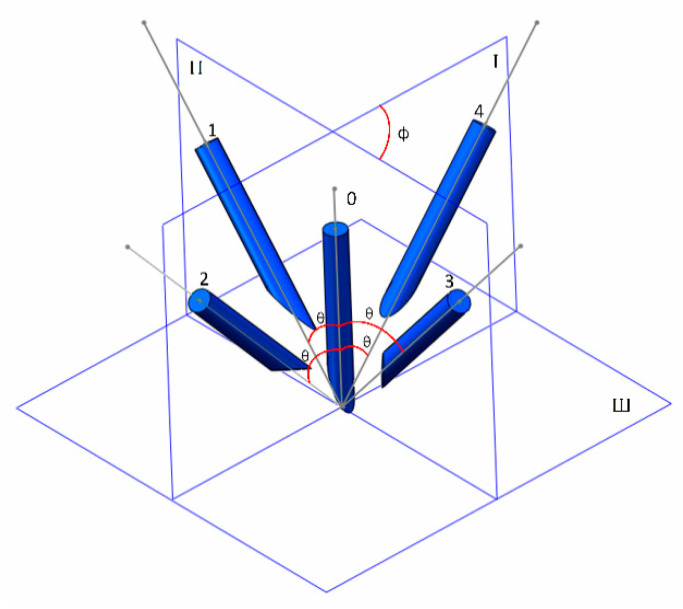
Configurations of TCP calibration.

**Figure 11 sensors-21-00366-f011:**
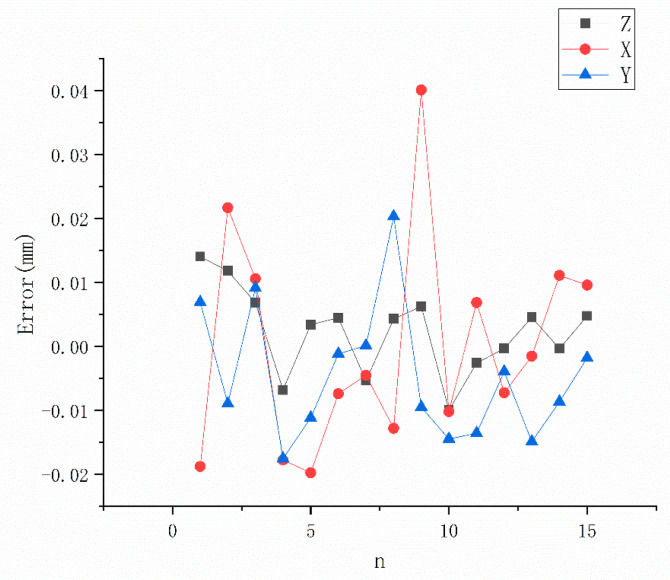
Measurement error of the vision system. (X, Y and Z are defined in the robot base frame).

**Figure 12 sensors-21-00366-f012:**
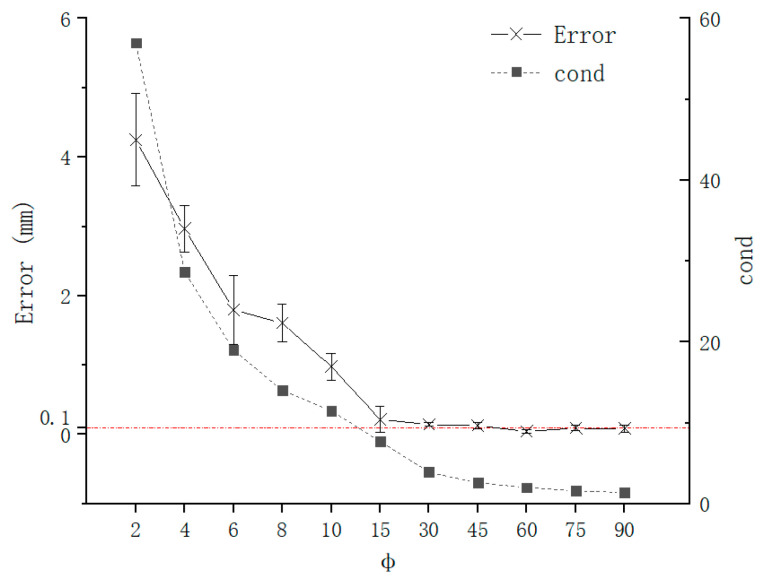
Error of TCPP with ϕ.

**Figure 13 sensors-21-00366-f013:**
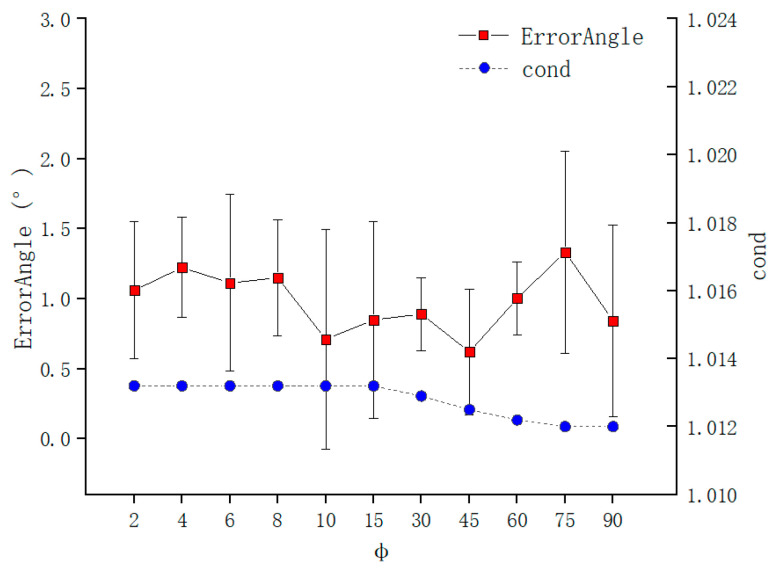
Error of TCPF with ϕ.

**Figure 14 sensors-21-00366-f014:**
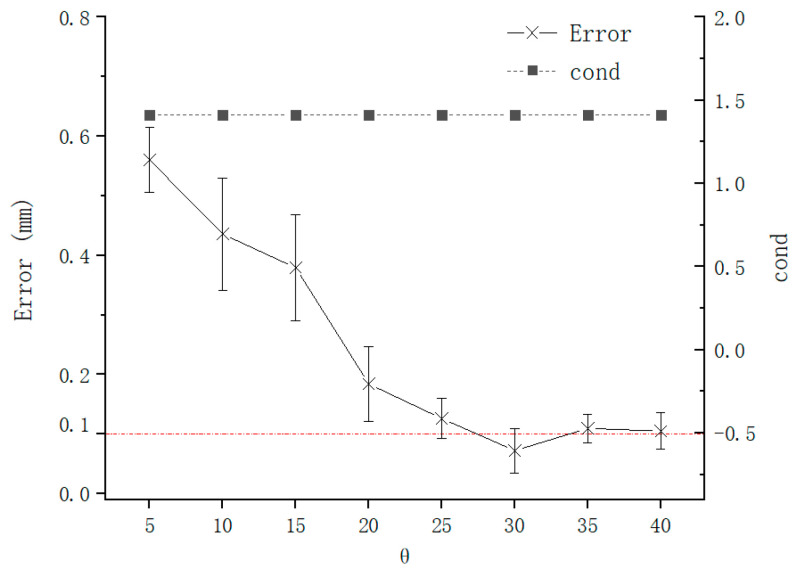
Error of TCPP with θ.

**Figure 15 sensors-21-00366-f015:**
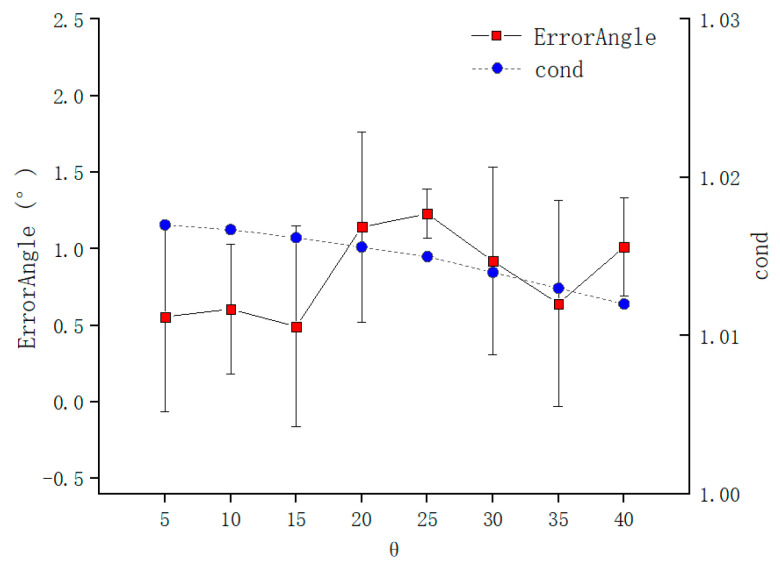
Error of TCPF with θ.

**Figure 16 sensors-21-00366-f016:**
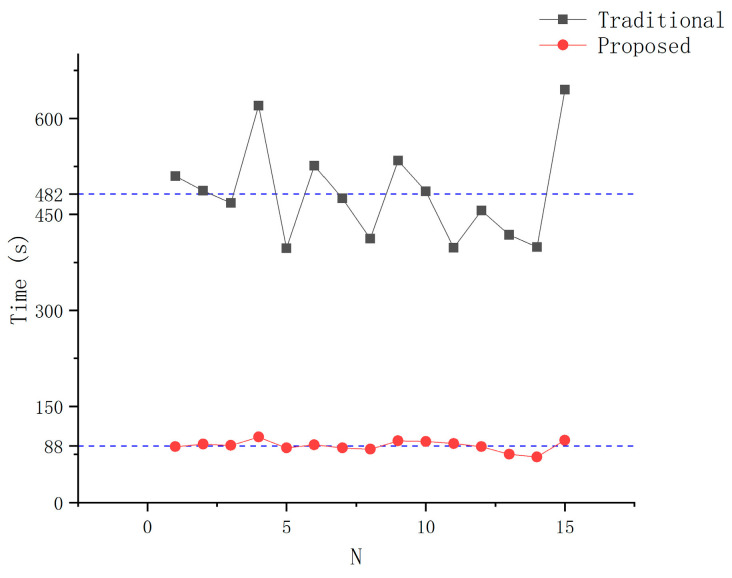
Time consumption.

**Figure 17 sensors-21-00366-f017:**
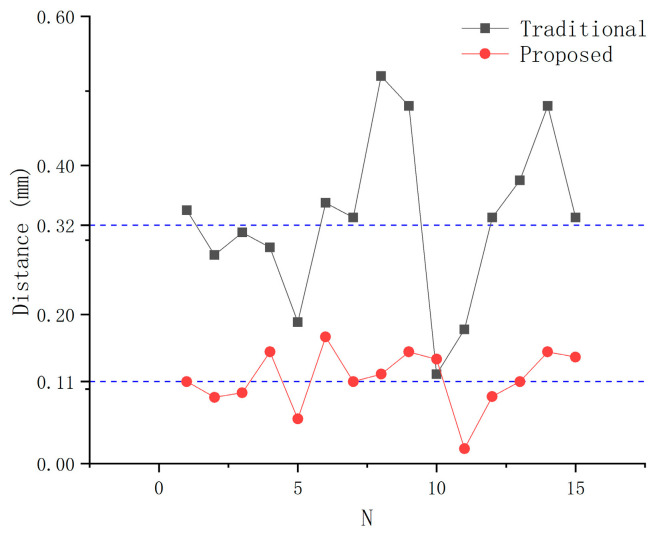
Comparison TCPP error.

**Figure 18 sensors-21-00366-f018:**
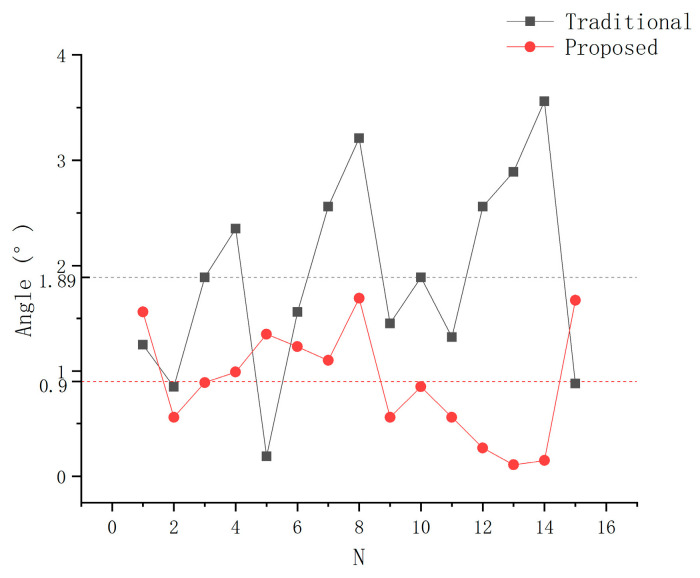
Comparison TCPF error.

**Table 1 sensors-21-00366-t001:** The details of configuration of experiment.

	ϕ																	
θ=40°	2°	4°		6°		8°	10°		15°		30°		45°	60°		75°		90°
	θ																	
ϕ=90°	5°		10°		15°			20°		25°		30°			35°		40°	

**Table 2 sensors-21-00366-t002:** The internal and external parameters of the vision system.

	Right Camera	Left Camera
Focus(mm)	12.39	12.41
Cell Width (Sx) (μm)	1.25	1.25
Cell Height (Sy) (μm)	1.25	1.25
Center Column (Cx) (pixel)	2387.07	2433.72
Center Row (Cy) (pixel)	1820.48	1861.79
2nd Order Radial Distortion (K1) (1/pixel^2^)	8.90 × 10^−10^	8.80 × 10^−10^
4th Order Radial Distortion (K2) (1/pixel^4^)	3.42 × 10^−17^	−1.06 × 10^−16^
6th Order Radial Distortion (K3) (1/pixel^6^)	−3.06 × 10^−24^	1.99 × 10^−23^
2nd Order Tangential Distortion (P1) (1/pixel^2^)	1.89 × 10^−13^	1.41 × 10^−13^
2nd Order Tangential Distortion (P2) (1/pixel^2^)	−1.56 × 10^−13^	1.56 × 10^−14^
Image Width (pixel)	4912	4912
Image Height (pixel)	3684	3684
Relative position (mm)	162.39, −4.7 × 10^−5^, 157.34	
Relative pose (°)	2.93, 270.23, 3.06	
Reprojection error (pixel)	0.28	

## Data Availability

The data presented in this study are available on request from the corresponding author.

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
