# Peer review of "Physician-Friendly Tool Center Point Calibration Method for Robot-Assisted Puncture Surgery"

_sensors, 2021, doi:10.3390/s21020366_

Round 1
Reviewer 1 Report
The current manuscript is an updated and extended version of an earlier submission. Authors corrected indicated mistakes and extended approach by considering orientation.
Paper presents an approach for vision supported calibration of tool centre point position and orientation with robotic application in surgery. Presented system uses two cameras of high resolution that monitor relatively small volume in which the TCP had to be manually jogged to. It is an improvement with respect to the traditional calibration method concerning accuracy and time consumption. However vision-based calibration approaches have been reported in past decades, also for surgery application.
Some specific comments:
Section 1/Abstract
By definition, TCP is a frame/transformation from robot base to TCP, not a physical entity, e.g. in the calibration procedure, operator is jogging tip of end-effector to the calibration tip to specify that TCP should be defined on that tip of end-effector.
Line53-54, description of orientation calibration of the TCP is not clear. If the orientation of TCP has to be specified then an additional 1 point on the Z-axis is needed and optional 1 point on X (e.g. of calibration procedure in industrial ABB robots). You need to be more specific about context – is it calibration procedure of UR5 robot?
Line 264 – justify that rotation about the needle axis is not relevant for the calibration. The tip of the needle is not symmetric and common sense indicate that its orientation is important while performing a task
Line 315: What is the accuracy of Laser tracker system used to measure the accuracy of the vision system
Section 3.3
Line 359 not clear statement: “operator controls the robot to move randomly six points along the XYZ axis separately under a fixed pose” 6 points in total, or 6 arbitrary points along each axis, in an arbitrary order or rather specified.
Entering the vision system’s view field should be the first step before orientation calibration.
Section 4.
Table 2. To be consistent with described procedure distortion parameters should be expressed in pixel domain as distortion removal is applied on image (pixel) level
Section 3.2, Eq. 16-17. With this approach, you assess the repeatability of calibration, not the accuracy. You might get each time very similar results but you do not know how close to real values those are. To assess absolute accuracy of a whole robotic system with calibrated TCP you could perform multiple tool reorientation and measure how the position of the needle tip is changing – the range of its motion indicate the accuracy of TCP calibration
Figure 11 (lines 371-181) indicate which error/formula is used (i.e. P_error/Eq15), X, Y, Z indicates the experiments where the robot was jogged along its corresponding axes or it is X, Y, Z coordinate of the difference (error). Negative values indicate that it is the second case. The absolute difference (distance) should be presented (P_error, as presented in eq 15 – x,y,z components could remain as additional information)
Discuss what are the components of measured error – e.g. Laser tracker, Vision system, robot (accuracy of the robot to keep constant orientation). If you want to check the accuracy of just vision system e.g. use object of known shape/dimensions and check the difference in obtained dimensions in the vision system.
Lines 371-181 Indicate what is the positioning error of used robot (spatial resolution and repeatability) and discuss how it affects presented calibration procedure (for camera and TCP calibrations).
Line 468 – there is: that theta should be as large as possible? Not precise, should it be 90, 180, 360 deg or infinity?
Symbol theta is used in sections 2 and 3 for different things that might be confusing e.g which theta is referred to in conclusions.
General comments:
Authors should better reflect on the design of the experiment and performed evaluations i.e. what is assessed when error measurements are performed in the proposed way, and how different components of the system might affect the final results.
Regarding orientation calibration, presented approach uses absolute orientation calibration of the vision system that in my opinion is not necessary – consume more time (it might increase accuracy as more points are used, but not sure). Calibration of TCP’s orientation could be done in a relative way i.e. relative orientation of TCP’s Z-axis (needle axis) and Z-axis of flange detected in vision system frame. It can be achieved by moving TCP in Z-axis of the flange frame(or any known direction in known robot frame) then obtain this direction in vision system frame and compare with detected needle orientation in the vision system.
It could have a higher value if the presented vision system was used for automatic calibration of TCP – automatic reorienting of the tool while keeping the same position of the needle tip (feedback control in image space, no calibration of vision system required)
Regarding the accuracy of the vision system, considering geometry and camera parameters 1 pixel corresponds to a spatial distance of 0.016 mm (theoretical resolution, with subpixel detections it can be increased)
Reviewer 2 Report
This paper describes a new calibration method of robot tool center point, which is operation-independent and easy to use for the operator. The text is divided into logical chapters and clearly describes the procedure of the proposed method. Unfortunately, the article is written in worse English with a number of missing punctuation marks and misused words. I recommend improving that.
I have a few recommendation notes:
28 - In the introduction chapter I miss the principle of how, according to the authors, robots are guided in medicine (if they were guided manually by surgeons, the inaccuracy of calibration below 1 mm does not matter; on the contrary, I cannot imagine any automatic guidance of the "vein tracking" type. Please describe the current state of robot guidance in medicine.
212 - It is not clear from the text how the needle tip (ie its TCP) is defined. It is important that when looking at the needle from the back, the image will not have the shape of fig. 6, but the tip will actually be U-shaped. Please precise this description to be clear.
262 - Figure 8 – Some of the coordinate systems are clockwise and some of them are counterclockwise, why? Unify the used coordinate systems.
278 - The sentence on line 278 is misleading, it should rather talk about a fixed rotation of the end effector.
283 - Equation (9) is based on the fact that the robot changes only the position and not the orientation of the tool. However, from a general point of view, this will only be the case with an absolutely accurate robot, which should be mentioned. Since you have a 3D tracker from Leica, you could relatively easily determine the geometric inaccuracies of the robot you are using.
315 - You declare the Leica tracker as a reference gauge, it would be appropriate to state the basic parameters of its accuracy.
382 - In general, in the development of errors (eg Figure 11), I would like to see a formulation of calculating that error. Here it is probably the difference between the length of the displacement measured by the Leica and the cameras, but what about the angles in Figure. 13?
403 - Finally, the same applies for the comparison of calibration errors. What is actually taken as the correct value from which the presented deviations are calculated?
Round 2
Reviewer 1 Report
Thank you for the corrections and explanations in the response.
The paper might benefit more if some of the aspects mentioned in the replay have been included in the paper e.g. one sentence about why authors “do not care” about needle rotation (as it is not negligence but a deliberate decision)
Regarding Figure 11 it will be beneficial to explicitly indicate that error is calculated according to formula (15) – just add a reference to (15) in line 385. Moreover, it should be better clarified what XYZ means in figure 11. From the response, I understand that X, Y, Z indicate errors (difference of distances and negative values are justified) from experiments where the robot was jogged respectively along X, Y, and Z direction of its base frame.
And final remark, the accuracy of laser tracker is 15um+6um/m that gives accuracy (with 2m distance to reflector) 27um = 0.027mm. It means that laser tracker error for distance measurement will be 0.054mm (0.027mm error for starting point plus 0.027mm for endpoint) as in error calculation always take the worst scenario. However, by repeating and averaging the measurement, the measurement error might be reduced. Recorded error, presented in figure 11, is a sum of laser tracker error and vision system. Because of that, I believe that the presented vision system might has better accuracy then indicated in the paper 0.05mm. But understand that to be on save side Authors overestimated the the system error.
Author Response
Please see the attachment.

This manuscript is a resubmission of an earlier submission. The following is a list of the peer review reports and author responses from that submission.
Round 1
Reviewer 1 Report
Paper presents an approach for vision-based assistance for calibration of tool centre point (TCP) with robotic application in surgery. Presented system uses two cameras of high resolution that monitor relatively small volume in which the TCP had to be manually jogged to. It is an improvement with respect to the traditional calibration method concerning accuracy and time consumption. However vision-based calibration approaches have been reported in past decades, also for surgery application. Because of this novelty and significance of presented work are limited. It could have a higher value if the presented vision system was used for automatic calibration of TCP including its orientation.
There are some flaws in the experiment and result presentation that cause it is hard to judge the value of the presented work. Equations should be carefully checked and way of result presentation should be improved.
Specific remarks:
- Potential mistakes:
- Line 244: should it be “transformation from B to V”?
- Line 246: should it be “transformation from B to Ei” ?
- Line 260 and 263, the wrong format of references to equations.
- Line 265, Equation 9: in first bracket indexes “Ei” and “Ei+1” should be exchanged with places or the right side of the equation should be multiplied by “-1”
- Line 313, Equation 12: is the last “-“correct? Equation presents the average value of error (that could be negative or positive) what hinder how spread and how big the errors could be. If authors want to use the average value as an estimator, it should be the average of absolute values.
- Line 318, Equation 13: continue the error from equation 9.
- Line 325, Table 1. The values of distortion seem to be odd. If the equations/method from referenced literature and library is applied, those should have values very close to 0.
- Figure 10: Does it use formula 12 for error calculation? It does not show how spread the error are. The maximal errors should be also presented for each experiment.
- Figure 12: Which equation is used? It is not the equation 13. If it is equation 12, then figure present accuracy of the vision system, not the accuracy of the calibration.
- Appearance improvements:
- Figure 2: provide a description of symbols (not all are indicated in the text e.g. B, omega) and values of parameters used in the simulation
- Figure 5: add scale, e.g. distance between holes in the mounting plate
- Lines 194-209 describe steps of processing. It would be beneficial to the reader to have a visualization of those steps. Consider moving enhanced figure 9 to this section or add additional figure illustrating processing steps. Consider changing the symbol for captured image i.e. “f” as it is also used for focal length in the next paragraph and this could be misleading.
- Line212,214: not all variables are described, are “u” and “v” the image coordinates after distortion removed? Used camera model should be presented in this place.
- Figure 6: visualize transformation between frames L and R as this figure is referenced also in line 217. Moreover, the orientation of frame R should be different from the orientation of frame L and be related to the orientation of camera R.
- Figure 7: Add the size of the calibration pattern.
Reviewer 2 Report
The manuscript describes a method based on computer vision techniques to identify and localize the tip of a surgery needle attached to a robot. The localization is based on traditional CV techniques with back-illumination, and a simplified stereovision camera location.
CONCERNS:
Bibliography is not completely appropriate, weakly referenced, and not well formatted. Claims seem to be exaggerated, CV based tools already achieve sub-millimiter accuracy (e.g. Tian, Yuan, et al. "Toward Autonomous Robotic Micro-Suturing using Optical Coherence Tomography Calibration and Path Planning." arXiv preprint arXiv:2002.00530, 2020;
Zhou, Mingchuan, et al. "Precision needle tip localization using optical coherence tomography images for subretinal injection." 2018 IEEE International Conference on Robotics and Automation (ICRA). IEEE, 2018;
Draelos, Mark, et al. "Optical Coherence Tomography Guided Robotic Needle Insertion for Deep Anterior Lamellar Keratoplasty." IEEE Transactions on Biomedical Engineering, 2019.)
Section 2.2 seems to be irrelevant theoretical analysis which is not linked to the following pose estimation algorithm. A better analysis can be performed by considering noise (and statistical) propagation on eq. (4). Even better if the authors refer to a classic work on the theme (e.g. Richard Hartley and Andrew Zisserman. Multiple view geometry in computer vision. Cambridge university press, 2003.)
Section 2.3
Many variables are used without explanation or the same names are reused to indicate different concepts. All the symbols in lines 195-202 were just introduced ones and never used later.
Overall the process is almost classic: cameras have been calibrated through an handmade procedure (both intrinsic and extrinsic parameters), then tip point is estimated through a second handmade triangulation.
Tip analysis is very simple, Hough transform, Hu moments, countour detection may provide more informative results. Background subtraction and Otsu compensation appear to be inappropriate for the computational pipeline. If the needle is black and the background is subtracted all converges towards the same color. Threshold quantization or color segmentation, may be more appropriate.
From line 212 to eq on line 214, the authors miss the typical unknown scale factor, which can be recovered only through stereography later. Moreover it is not clear from those eqs. why SxL and SyL should differ (pixels are squared).
\rho in e.3 is not introduced. Scale factor is already kept into account as 'z' there. Moreover since the product of the three terms on the right side has one in the last row, \rho can only be one.
r1--r9 in eq. (3) belongs to SO(3) so the solution in eq. (4) shall depend only from 6 parameters not 12.
Solution in eq (4) is much well described in traditional stereo-imaging books (e.g. Zielinsky) and makes use of projection matrices and/or the fundamental matrix.
Section 2.3
Procedure in steps (7), (8), (9) is addressed in classic robot identification textbooks. The authors here forget to estimate the minimum amount of pose variability which is adequate for a proper EndEffector-ToolTip transformation identification.
Terminology has completely changed in this section and really confusing for the reader. Translation is now indicated with capital (M), while before "t" was used. Lest camera is V(L) with T_V^B transform (why V). Right camera is only R, transform matrix is not indicated.
On line 244 the authors claim that transformation among BE is unknown during the calibration, but it should be known through the robot direct kinematics. They are T_V(L)^B and T_P^E to be unknown but constant. On line 246 letter 'V' should be 'E', and accordint to the arrow the transformation is the opposed one.
Section 3
It is unclear how the Leica system was used to detect exactly the needle tip. All the experiment refer to differential motion so it is unclear how absolute accuracy could be estimated is no accurate markers where placed on the Robot EE.
Section 4
Errors should be measured in modulus on all components not on individual ones. Being differential estimates these errors refers to robot sensitivity not to overall accuracy. Moreover the motion procedure was not described (sequence of points and orientations), but it can greatly influence the results (e.g. pure traslation motions only estimates for the robot accuracy).
Final remark,
when combining accurate kinematic motion from robot with camera data, one camera is enough (i.e. you do not need stereo vision) as done in most of existing work in literature. Stereo-vision is useful only if it eliminates calibration.